# The Compass Error Comparison of an Onboard Standard Gyrocompass, Fiber-Optic Gyrocompass (FOG) and Satellite Compass

**DOI:** 10.3390/s19081942

**Published:** 2019-04-25

**Authors:** Krzysztof Jaskólski, Andrzej Felski, Paweł Piskur

**Affiliations:** 1Polish Naval Academy, Institute of Navigation and Maritime Hydrography, Smidowicza St, 69, 81127 Gdynia, Poland; a.felski@amw.gdynia.pl; 2Polish Naval Academy, Institute of Electrical Engineering and Automatics, Smidowicza St, 69, 81127 Gdynia, Poland; p.piskur@amw.gdynia.pl

**Keywords:** gyrocompass, accuracy, fiber-optic gyrocompass, satellite compass, band-stop, finite impulse response (FIR) filter, spectrum analysis, Fourier Transform

## Abstract

The aim of the presented research was to analyze the accuracy indications of three types of compass systems for the purposes of meeting warship modernization requirements. The authors of this paper have made an attempt to compare the accuracy of an onboard standard gyrocompass, a fiber-optic gyrocompass (FOG) and a satellite compass in real shipping circumstances. The research was carried out in the Gulf of Gdansk area, during the preparation of hydrographic surveys on stable courses. Three heading recordings have been taken into consideration. The helmsman’s operation and vessel inertia were analyzed and removed according to a spectrum analysis. Transient characteristics and the spectrum analysis (based on the Fourier transform theory and headings descriptions in the frequency domain) are presented. Data, processed using a band-stop finite impulse response (FIR) filter to reduce low-frequency heading distortions, are presented for further analyses. The statistics of errors of the compasses investigated, as well as the spectrum of these errors, are also presented. Based on accuracy measurements, the possibility of using the most accurate heading data as the input signal to the automatic ship control system was considered.

## 1. Introduction

Despite being equipped with receivers of the Global Navigation Satellite System (GNSS), Automatic Radar Plotting Aid (ARPA), and Electronic Chart Display and Information System (ECDIS), contemporary ships still cannot sail without a compass. According to the International Convention for the Safety of Life at Sea (SOLAS) convention and International Maritime Organization (IMO) regulations, a gyrocompass is obligatory. A magnetic compass (or simply compass, according to IMO terminology) is based on a 19th-century design, but nowadays is equipped with elements assuring transmission to contemporary digital devices. Interestingly, gyrocompasses possess a number of similar limitations to their predecessor, except for the many advances made in the form of digital elements. However, more advanced solutions have been implemented over the last 100 years [1]. In the 1980s, laser gyrocompasses were introduced on warships, followed by fiber-optic modifications in the 1990s. The latter have become increasingly popular, even on small, unmanned vehicles [2,3], and have also been accepted by off-shore suppliers and hydrographic ships.

Satellite compasses trace in a completely new way [4,5]. It is a specific version of a GNSS receiver having, inter alia, an option to determine a ship’s heading. These devices were accepted by many users, but current regulations do not allow the use of them as a substitution for a ship’s compass.

It is evident that different construction solutions possess different metrological characteristics. The gyrosphere of the classical gyrocompass behaves as the pendulum, meaning it possesses its own oscillations. These oscillations appear after maneuvers of the ship, especially after rapid alterations of the heading. However, present compasses, except for other modifications, are characterized by miniaturization, so one can expect less influence of the gyroscopic sphere inertia on the components of the instrument. Authors did not find such information in the literature.

Alternatively, fiber-optic gyrocompasses (FOGs) and satellite compasses do not possess dynamic elements, so one can expect a different nature of system response. It is especially difficult to find information on the behavior of these new constructions, which are installed on typical ships, in the literature. Additionally, it should be taken into account that the dynamics of the reaction of a ship is dependent on her maneuverability and potential to maintain her on a stable course, which can only be shown by the onboard installed compass. Thus, the dynamics of the indications of compasses are relative to their properties, but also to real reactions of the ship, and develop according to the value of the heading shown by the compass in use.

In literature, this problem is treated superficially; usually, the accuracy of a compass is given along with its possible deviation. The most common approach to the problem is based on the foundation of the random character of errors and is defined by a standard deviation. It is worth mentioning that we have no evidence that the process has a normal distribution. However, many uses in the area of sensor integration demand familiarity with the spectrum of errors. Over the last 100 years, many research papers [4,5] concerned with the nature of errors of the gyroscopic compass have been published and it is known that these errors had a character of the ergodic process with low frequencies. For certain, we know less about the character of errors of more advanced solutions. Some investigations in this field have been executed by the Polish Naval Academy and have been published [6], but these investigations have been devoted to installations with mechanical gyroscopes, which are currently not in use. More complex analysis has been conducted and documented in Reference [1], with the results of similar studies on satellite compasses being published in Reference [4] and Reference [5]. Interesting research outcomes of preliminary numerical simulation studies and the preliminary static experimental evaluation of a true-North gyrocompass system—employing a commercially available low-cost inertial measurement unit (IMU) comprising a 3-axis FOG with micro-electro-mechanical systems (MEMS) accelerometers—were presented in [7]. The article presents the results of a numerical simulation study to evaluate the potential accuracy of the proposed system, and a numerical sensitivity study to evaluate how this accuracy will change with variation in the sensor measurement noise of the gyroscopes and accelerometers. Interesting results were achieved in Reference [8] with the use of a MEMS gyroscope, suitable for northfinding in pointing and targeting applications with the bias stability of 0.03 deg./h.

Similar accuracy analyses have already been presented in recent years, however, they were limited to single devices [9,10,11] and have been focused on small autonomous vehicles [2]. Some aspects of this problem have been discussed in books concerning the integration of global positioning systems (GPSs) and inertial techniques, for example Reference [12] or Reference [13]. Many ship accidents have arisen from an error in course indication. Considering that actual errors in gyrocompasses and satellite compasses are very minor, such devices may be regarded as valid inputs for autopilot functions, provided that any failure in their operations is controlled by means of a secondary heading source [14]. Meanwhile, real information regarding both the character of the oscillations and the dynamics of a compass in real circumstances encounter the problem of separating the natural oscillations of the ship (yaw) from the oscillations of the compass, resulting from its design. That is why the results of an investigation of any devices devoted to robotics cannot be translated to a ship’s devices, as the latter are designed with different tasks and operate in different circumstances.

Research material collected during such experiments as those discussed above can be used to analyze the effect of a ship’s motion on the accuracy of its compass readings. It was assumed that the main disturbance—having an essential significance with regard to the accuracy of the readings of a compass—would be the ship’s motion.

It should also be remembered that the real ship’s heading deviations would result from the movement of the ship. In other words, the motion will determine both the real ship’s heading deviation and the errors in measuring it.

It might seem that the comparison between the compass headings with the reference heading, determined by Equation (1), will enable a full analysis of changes caused by the ship’s motion. Unfortunately, it appears that this kind of comparison will only provide information on absolute values of heading deviations for specific moments of observation, giving no information on their character. The analysis conducted led to the conclusion that it is necessary to analyze the distributions not only in the time domain but also in the frequency domain. One of the basic tools used in the theory of signal processing is Fourier transformation, which was used for frequency domain analysis.

This paper is organized as follows: Section 2 discusses the measuring principles and the design of experiments. Based on the measurements obtained at time interval k=5 s, the heading recordings of three types of compass systems are presented. Section 3 describes the transient characteristics and spectrum analyses that were carried out after heading reduction with the use of MATLAB R.2015 software. Harmonic components resulting from long periods of disturbances by the helmsman and ship inertial possibilities were attenuated. Section 4 provides discussion of the method and results. Finally, conclusions are presented in Section 5.

The research questions this paper attempts to answer are as follows:

Does the model of the band-stop finite impulse response (FIR) filter eliminate low frequency heading distortions?

What frequencies should be used to filter out the harmonic component resulting from long periods of disturbances by the helmsman?

The output signal of which compass is the most accurate and has the most potential for use as an autopilot input signal?

## 2. Materials and Methods

### 2.1. Background

According to their technical specifications, the satellite compass, gyrocompass and FOG compass transmit National Marine Electronics Association Interface Standard for communication between marine electronic devices (NMEA) 0183 messages with the HDG (heading gyro) or HDT (heading true) sentence. The station for signals recording consisted of a FOG NAVIGAT 3000 and a satellite compass FURUNO SC50, shown in Figure 1, which were developed at the Polish Naval Academy. These were installed on the survey vessel ORP “HEWELIUSZ”.

The investigation was carried out in the area of the Gulf of Gdansk, during hydrographic surveys from the 8th to 9th of February, 2018, with the course over ground 091.5 deg. and 271.5 deg. according to the directions of the hydrographic profiles.

At least ten data recordings were carried out. For the first four data recordings, the time of the measurements was originally 200 s, but the first calculation indicated the dominant frequencies equal to 0.01 Hz. Thus, the recording time was extended to no less than 1000 s and the sampling frequency of recorded data was changed from f(s)=1 Hz to f(s)=0.2 Hz. The weather conditions during the test numbers 4, 6 and 7 were: Wind SWbW-2B (Beaufort) and state of the sea 1.

According to the technical specifications of the navigational instruments, the accuracy of NAVIGAT FOG 3000 was [15]:$$\delta_{FOG}=0.8{}^{\circ}\cdot \sec\left(\varphi_{m}\right),$$
while the accuracy of the satellite compass FURUNO SC50 was [16]:$$\delta_{SC50}=0.5{}^{\circ}\cdot \sec\left(\varphi_{m}\right).$$

The reference gyrocompass, installed on board and presented in Figure 2, was NAVIGAT X MK1, with an accuracy of [17]:$$\delta_{MK1}=0.4{}^{\circ}\cdot \sec\left(\varphi_{m}\right).$$
where $\varphi_{m}$ is the mean geographical latitude.

As the mean geographical latitude of the area was 54.6 deg. N, the mean error of each device could be determined. The results of such calculations are shown in Table 1.

### 2.2. The Scope of Research

To fully and explicitly determine the error values, as well as the degree of their dependence on the vessel’s motion, it was necessary to compare the directions measured with a reference direction. Discussion concerning the variation of these errors cannot be described in a deterministic way. One of the fundamental problems related to the conduct of investigations of compass errors is the selection of a reference heading. A reference heading can be used for comparison against the heading defined by the compasses under investigation. In the current investigation, the reference heading was calculated as the mean of all the devices, according to the formula:(1)$$RH=\frac{1}{m}\left[\begin{array}{c} \frac{\sum_{i=1}^{n}\left(CH{\left(i\right)}_{FOG}-\frac{\left(\sum_{i=1}^{n}CH_{FOG}\right)}{n}\right)}{n}+\frac{\sum_{i=1}^{n}\left(CH{\left(i\right)}_{SC50}-\frac{\left(\sum_{i=1}^{n}CH_{SC50}\right)}{n}\right)}{n}\\ +\frac{\sum_{i=1}^{n}\left(CH{\left(i\right)}_{MK1}-\frac{\left(\sum_{i=1}^{n}CH_{MK1}\right)}{n}\right)}{n} \end{array}\right],$$
where RH is the reference heading, CD(i) is the course distortion for the iteration (i), $CH{\left(i\right)}_{FOG}$ is the compass heading per FOG compass for the iteration (i), $CH{\left(i\right)}_{SC50}$ is the compass heading per satellite compass for the iteration (i), $CH{\left(i\right)}_{MK1}$ is the compass heading per standard gyrocompass for the iteration (i), n is the number of iterations, and m is the number of compass systems. Note that in this study m=3.

In References [9,18,19], comparisons between the direction defined and the direction of the reference system for accuracy determination were presented. In these works, the satellite systems were used to determine the heading and an estimation of the course system accuracy. The methods performed consisted of comparing the defined direction with the direction of the survey profile, which may be used to check the accuracy of a shipboard compass for constant errors. Excluding the fact that the authors reduced the accuracy and were dependent on many factors including visibility, wind speed, currents, state of the sea, and helmsman’s ability to keep the ship accurately in line with the survey profile, their research provides a comparison for short periods of observation. Consequently, the methods described could only be used to determine constant correction, called compass error (CE), but could not investigate compass accuracy in dynamic conditions.

In the current experiment, recordings were made based on constant heading readings from a classical gyrocompass with internal correction, a FOG compass and a satellite compass. Exordial, short data registrations lasting up to 200–350 s were discarded, as they contained too little information. For further discussion in this paper, heading recording numbers 4, 6, and 7 were taken into account as the most valuable. The heading recordings from the three types of compasses as a function of time were presented in Figure 3, Figure 4 and Figure 5.

Analyzing the heading distributions, presented in Figure 3, Figure 4 and Figure 5, periodical deviations mainly caused by yawing can be identified. The difference in amplitudes and the phases of these deviations shows the varying character of both distributions resulting from the gyro element suppression of oscillation for the classical gyrocompass. A satellite compass FURUNO SC 50 equipped with MEMS (micro-electro-mechanical systems) technology is more resistant to short-period swinging than a gyrocompass NAVIGAT X MK1. The course distortions depends on the potential to set a time of smoothing function based on the Kalman filtering methodology [3].

These examples show perceptible resemblances of the oscillation in all indications of the three compasses. Certainly, this is the result of the ship’s yaw, which is especially visible in Figure 4 and Figure 5. However, in Figure 5, the gradual approach of values recorded from NAVIGAT X MK1 to values shown by the remaining systems can be observed. This is the characteristic behavior of the classical gyrocompass after alteration in the ship’s course.

In Figure 4 and Figure 5 the satellite compass and FOG indications are analogous, even in terms of the phase, while in Figure 3 the phase shift in SC 50 indications—in relation to the reference system (NAVIGAT X MK 1)—can be observed. Differences are even up to 3 degrees, while the accuracy of each instruments amounts approximately 1 degree. During the experiment the ship navigated with meridional (E and W) courses, so speed deviation, as a possible additional source of error, did not appear.

## 3. Results

### 3.1. Transient Characteristics and Spectrum Analysis

Three data recordings with the use of post-processing methodology were taken into consideration. The effect of ship motion and the accuracy of the three compass systems were analyzed. In addition, the effects of the helmsman’s operation and vessel inertia were analyzed and removed with the use of spectrum analysis.

Firstly, course distortions CD(i,j) were calculated according to the formula:(2)$$CD\left(i,j\right)=\left[\left(CH\left(i,j\right)-\frac{\left(\sum_{i=1}^{n}CH_{j}\right)}{n}\right)\right],$$
where CD(i,j) is the course distortion for the iteration (i), and the compass type (j) (FURUNO SC 50, or NAVIGAT X MK1, or NAVIGAT FOG 3000), $CH_{j}$ is the compass heading according to instrument type, and n is the number of iterations.

The course distortions results are presented in Figure 6, Figure 7 and Figure 8.

Secondly, absolute deviation outcomes for heading recordings No. 4, 6, and 7 are presented in Table 2.

The main disturbances in heading are not only the result of the ship’s unstable motion, (changes in heading). Therefore, the ship’s motion was determined by the heading deviation (dynamic properties of the ship) and at the same time measurement errors of heading deviations were generated by the ship regardless of whether it was being manually controlled or in automatic mode. The comparison of the compass heading with the reference heading was determined by the onboard gyrocompass, according to the reference heading calculations and hydrographic profile direction. The comparison delivers information about heading deviations for the moments of observations according to gyrocompass signal recordings without information about their character of changes. The recorded assumptions were presented as follows:If:Δt=5 s,then:f(s)=0.2 Hz
where f(s) is the sampling frequency and Δt is the sample period.

The recorded assumptions depended on the errors which appeared in the classical and FOG gyrocompasses. Errors caus a long period deviation, which in itself depended on the dynamic deviation.

In conclusion, the analysis of the distribution in the time domain was insufficient. Thus, it was necessary to analyze the distribution in the frequency domain. For this purpose, a basic tool in signal processing theory—the Fourier transform theory—was applied. This tool ensured a signal description determined in the frequency domain f or pulsation ω=2πf.

Signal x(t) can be presented according to the Fourier integration formula [6,20,21]:(3)$$x\left(t\right)=\frac{1}{2\pi }\int_{-\infty }^{+\infty }X(\omega )\cdot e^{j\omega t}d\omega ,$$
where X(ω) is the Fourier transform of signal x(t).

The Fourier transform of signal x(t) is as follows [6,20,21]:(4)$$X\left(\omega \right)=\int_{-\infty }^{+\infty }x\left(t\right)\cdot e^{-j\omega t}dt,$$

Contrary to the continuous signal, x(t), the distribution of heading is determined by the discrete signal of the periodical character x(i). The period of sampling for such a signal is given by:(5)x(i)=x(iΔt),
where *i* = 0 … *N*−1.

Mathematical notation of the signal is [6,20,21]:(6)$$x\left(i\right)=A\left(0\right)+\sum_{r=1}^{\frac{N-1}{2}}A\left(r\right)\cdot \cos\left[\omega^{\prime }ri\cdot t+\varphi \left(r\right)\right],$$
where A(0) is the initial signal amplitude, A(r) is the amplitude of the harmonic signal components, $\omega^{\prime }$ is the pulsation of the main components for the harmonic signal, where r=1, and φ(r) is the phase of the specific harmonic signal component.

The pulsation of the main components for the harmonic signal, where *r* = 1, is [6,20,21]:(7)$$\omega^{\prime }=\frac{2\pi }{N\Delta t},$$

The notation (6) of the signal x(i) gives information about what part of the distribution x(i) has the particular components in the form r for the harmonic signals of amplitudes A(r), phases φ(r) and initial amplitude A(0). The notation is true for odd N. For even N, r has to be in the range of $1\ldots \frac{N}{2}$ [6,20,21]. Note that the relationship was found to exist between the evenness of the samples, and the mathematical notations were presented in odd N elements mathematical apparatus.

Thus, Fourier analysis of the determined signal (6) enabled the distribution of the frequency domain. The amplitude frequency of signal A(s) is constrained in the lower range of frequency $f^{\prime }$ determined with the observation period T in the upper range of Nyquist frequency f(n), where [6,20,21]:(8)$$f\left(n\right)=\frac{f\left(s\right)}{2},$$
where f(s) is the sampling frequency.

The main task of the mathematical model for the ship’s heading distribution was finding an amplitude of heading errors. It is more convenient to study the complex spectrum. This is the heading distribution and the heading disturbances distribution.

According to equation:(9)CE(iΔt)=CH(iΔt)−RH,
where CE(iΔt) is the heading disturbances for the sample i=0…N−1, RH is the reference heading according to Equation (1), and CH(iΔt) is the compass heading for the sample, i.

The compass heading was maintained according to the gyrocompass readings, with long periods of disturbances by the helmsman being implemented, as presented in Figure 3, Figure 4 and Figure 5. Before frequency analysis, the calculated RH (1) had to be reduced by subtracting the recorded compass headings according to formula CE(iΔt) (9).

The post-processing data for the FURUNO SC50, NAVIGAT FOG 3000, and NAVIGAT X MK1 after heading reduction (000 deg.) are depicted in Figure 9, Figure 10 and Figure 11.

Frequency analysis for compass course distortions depends on long-term disturbances. The distributions of compass heading distortions were equally susceptible to disturbances for the satellite compass, FOG compass and classical gyrocompass, and such disturbances depend on ship motion noises. The frequency analyses for compass course distortions are presented in Figure 12, Figure 13 and Figure 14.

The single-sided amplitude spectrum of three types of compasses for the heading recording No. 4, depicted in Figure 12, presents a harmonic partial of 0.015 Hz for the course distortions, with a maximum amplitude spectrum of 1.3 deg. These course distortions depend on the implementation of long periods of helmsman/autopilot disturbances and the ship’s inertial abilities.

Similar investigations were carried out with the use of heading recording No. 6. The single-sided amplitude spectrum, presented in Figure 13, depicts a harmonic partial of 0.01 Hz for the course distortions, with maximum amplitude spectrum of 0.9 deg. As in the previous case, the greatest course distortions depended on the disturbances created by the helmsman/autopilot and the ship’s inertial abilities.

The same research results on the basis of heading recording No. 7 registered data, presented in Figure 14, exhibit a harmonic partial of 0.01 Hz and an amplitude spectrum of 0.68 deg.

The low-frequency distortions below f=0.02 Hz are a phenomenon of undetermined fluctuations. This phenomenon is related to Schuler oscillations for the frequency distortions f=0.0018 Hz, presented in Figure 12, Figure 13 and Figure 14. This oscillations make a standard compass NAVIGAT X MK1 heading deviation from 0.5 deg. to 0.65 deg., which was a source of heading for the helmsman. If the helmsman had diverted the course according to the standard gyrocompass, the course distortions would have been duplicated by the FOG and satellite compasses.

### 3.2. The Model of the Band-Stop Filter with the Finite Impulse Response (FIR)

One solution for disabling low frequency heading distortions is to employ the FIR model presented in Figure 15.

where:

f(s)=0.2 Hz, the sampling frequency,

f(pass1)=0.005 Hz, the first passband frequency,

f(stop1)=0.0075 Hz, the first stopband frequency,

f(stop2)=0.0125 Hz, the second stopband frequency,

f(pass2)=0.015 Hz, the second passband frequency.

According to the filter model assumptions, and the results of the single-sided amplitude spectrum (presented in Figure 12, Figure 13 and Figure 14), the bandwidth from f(pass1)=5 mHz to f(pass2)=15 mHz will be attenuated, and the other frequencies will be passed. Band-stop FIR was proposed to eliminate the helmsman’s distortions. The helmsman’s disturbances frequency could not be measured directly, and hence filter parameters were estimated according to the system’s accuracy investigation and the analyzed spectrum results. With the use of a band-stop finite impulse response filter, it is possible to filter out harmonic components resulting from long period helmsman’s disturbances and ship inertia possibilities. The recorded signal and the signal after filtration are presented in Figure 16, Figure 17 and Figure 18.

According to the research outcomes presented in Figure 16, Figure 17 and Figure 18, the heading amplitude for the filtered data was reduced. Additionally, the harmonic component resulting from long periods of disturbances by the helmsman and the ship’s inertial possibilities were attenuated.

According to the research outcomes for the filtered data presented in Figure 19, Figure 20 and Figure 21, the heading amplitude (calculated according to Equation (2)) was halved. This is easily observed by comparing Figure 6, Figure 7 and Figure 8 with Figure 19, Figure 20 and Figure 21, respectively.

The band-stop finite impulse response filter halved the amplitude of the heading distortions and reduced the mean errors for the three heading recordings. The research outcomes are presented in Table 3 and Table 4.

Mean errors of the investigated compass systems (illustrated in Table 1 and Table 4) indicate a double reduction of potential errors after FIR usage, which results in the three compass systems having similar accuracy. Similar conclusions can be drawn after analyzing the compass heading mean distortion outcomes.

## 4. Discussion

The comparison of three compass systems, FURUNO SC 50, NAVIGAT X MK1, and NAVIGAT FOG 3000, was elaborated in both the time and frequency domains. Investigations were conducted during bathymetric works executed by the survey ship ORP “HEWELIUSZ”. Measurements for data analysis were taken when the ship navigated in the directions of nearly E and W, which was beneficial for the diminution of errors for all instruments. Examples of the registration (Figure 3, Figure 4 and Figure 5) indicated that the ship’s direction of movement had no impact. The registrations also appeared to suggest that there were no essential differences in the accuracy of the examined compasses; although the uncertainties (errors) were different between consecutive tests (around 0.3 deg.), the differences between the heading recordings from the compasses in the same test were similar.

The low-frequency distortions were in the band of 0.01 Hz<f<0.02 Hz This could be related to the Schuler oscillations of the NAVIGAT X MK1 compass, which was the source of information for the helmsman and—as consequence—was duplicated by the remaining compasses as the ship, in fact, navigated along such courses.

Analysis of the courses taken in the time domain gave the opportunity to diagnose the problem deeply and to eliminate the helmsman’s impact on the gyrocompass’ errors. From this experiment, we established that the uncertainty (errors) of the three examined instruments were similar. It was found to be approximately 1 to 1.5 deg., but it must be stressed that the experiment took place in good weather conditions, which was a requirement for the bathymetric measurements to be taken. Owing to the ship’s movement on a stable direction for the bathymetric profiles, a specific methodology for calculating the reference direction was proposed.

The Fourier transform was used for more accurate analysis. The frequency domain analysis of the ship’s motion distortions creates a new research opportunity with respect to the ship’s inertia, and weather conditions impact, etc. The frequency analysis of the compass errors constitutes a useful tool for investigating a simultaneous data error comparison. Frequency analysis enables the isolation of errors according to the ship’s yawing. This errors appear in data recordings regardless of the compass type. Frequency analysis creates an opportunity for analysis of the outer factors of the measurement process and provides a better methodology for compass error estimation. This method could be very important in estimating errors of navigational devices. With the use of a band-stop finite impulse response filter, it is possible to filter out the harmonic components resulting from long periods of disturbances by the helmsman and a ship’s inertia possibilities.

## 5. Conclusions

This paper discusses the accuracy of three different ships compasses. On the basis of parallel recordings made by the different devices installed on the same ship, the authors were able to investigate the uncertainty of measurements in the context of both amplitude and frequency. In the real experiment, it was checked that the price indications of different compasses differed insignificantly.

The frequency analysis of compass errors constitutes a useful tool for investigating simultaneous data error comparison. It was demonstrated that frequency analysis enables the isolation of the ship’s yawing errors. The model of the band-stop FIR is a good tool for eliminating low-frequency heading distortions. According to the filter model assumptions, and the results of a single-sided amplitude spectrum, the bandwidth from f(pass1)=5 mHz to f(pass2)=15 mHz will be attenuated and other frequencies will be passed. Band-stop FIR was proposed to eliminate helmsman distortions. After band-stop FIR usage, the accuracy of the three types of compass systems appeared to be comparable. During the experiment, the ship navigated according to information from a NAVIGAT X MK1 classical gyrocompass. Consequently, a low-frequency component existed in each registration, probably being descended from the dynamic properties of this onboard compass. The association between the dynamic properties of the ship and the properties of the gyrocompass is interesting. If a different compass—without this low frequency oscillation—was the source of information for the heading, how would the remaining systems of the ship behave? This is a potential direction for future investigations.

## Figures and Tables

**Figure 1 sensors-19-01942-f001:**
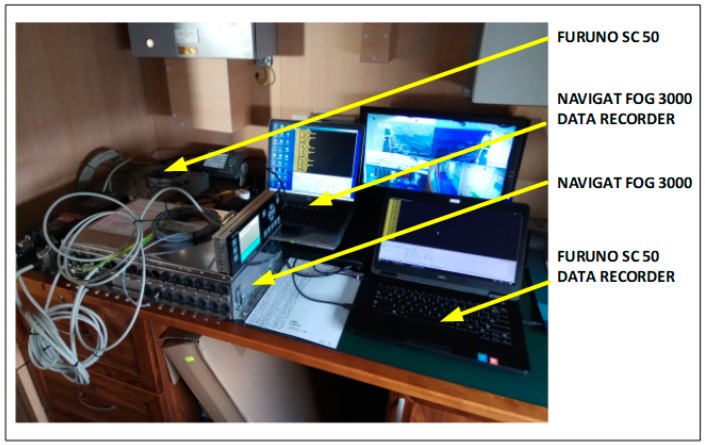
NAVIGAT FOG 3000 and satellite compass FURUNO SC50 recording stations (source: K. Jaskólski).

**Figure 2 sensors-19-01942-f002:**
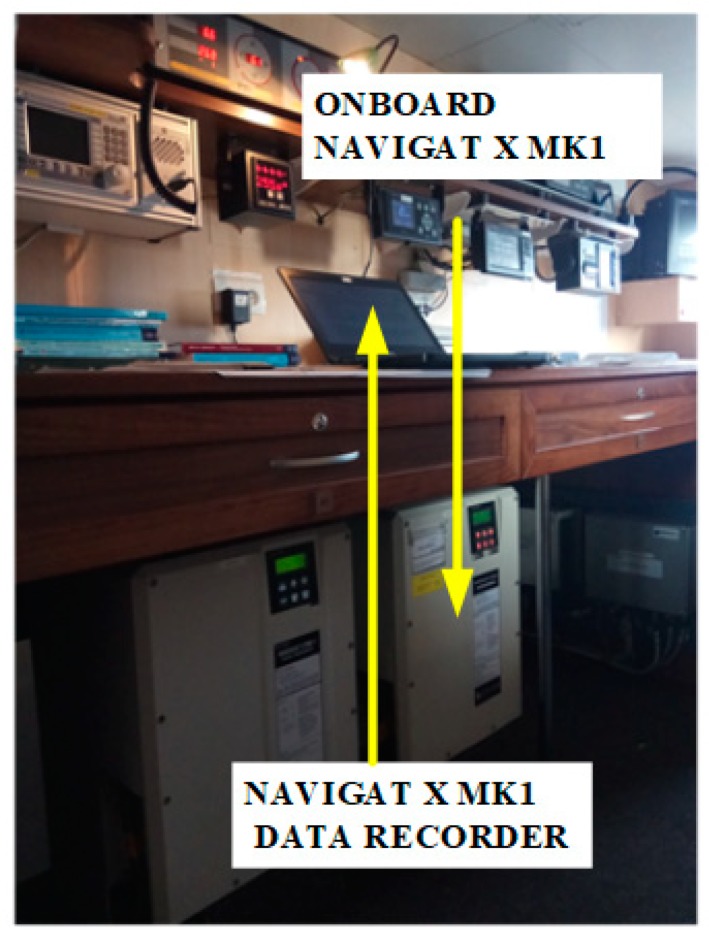
Shipboard gyrocompass NAVIGAT X MK1 as a recording station (source: K. Jaskólski).

**Figure 3 sensors-19-01942-f003:**
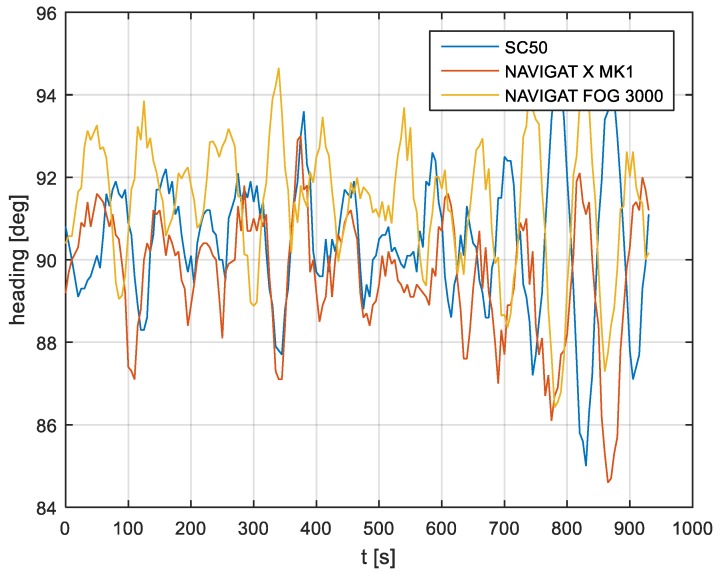
Heading recording No. 4: Measured data before data processing.

**Figure 4 sensors-19-01942-f004:**
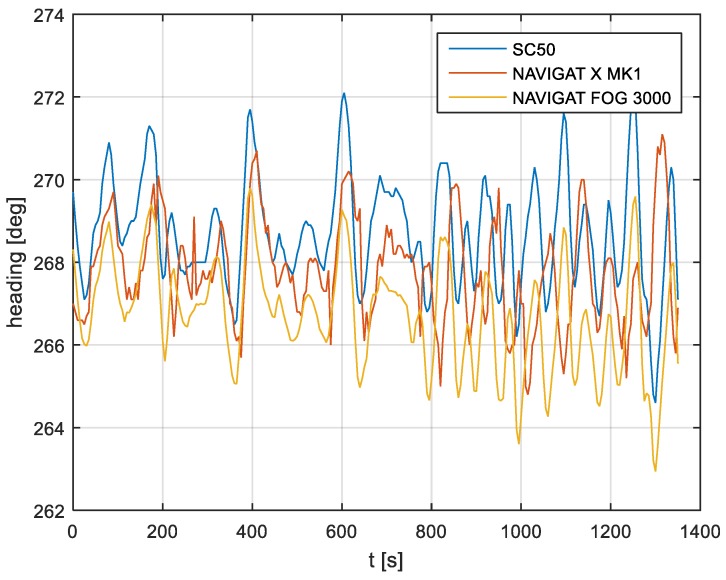
Heading recording No. 6: Measured data before data processing.

**Figure 5 sensors-19-01942-f005:**
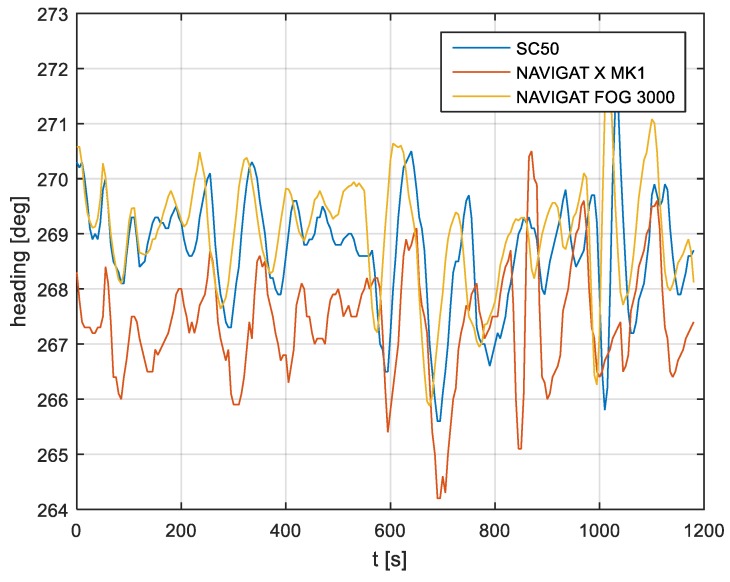
Heading recording No. 7: Measured data before data processing.

**Figure 6 sensors-19-01942-f006:**
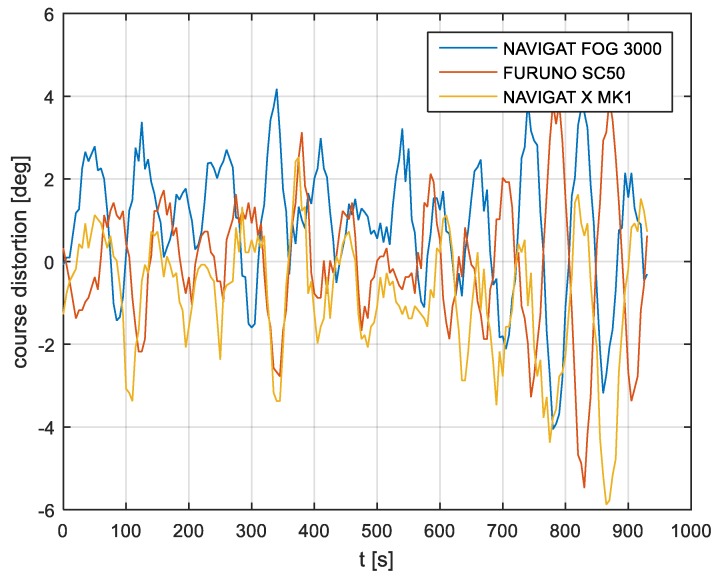
Heading recording No. 4: Course distortions.

**Figure 7 sensors-19-01942-f007:**
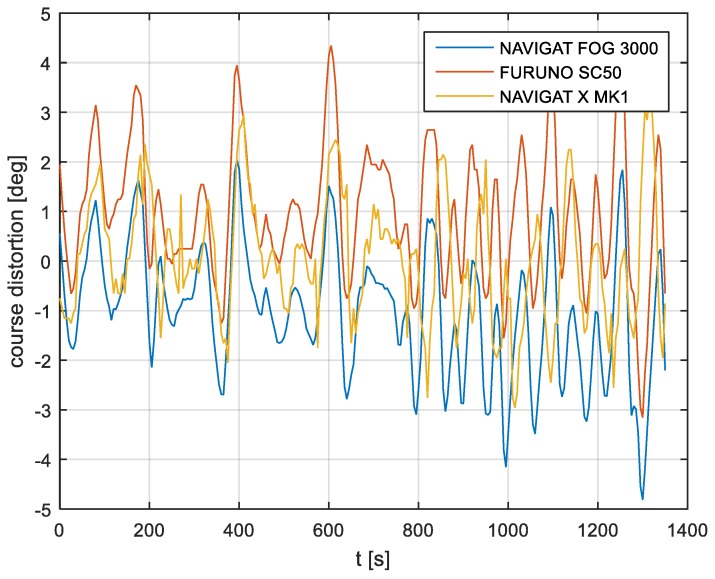
Heading recording No. 6: Course distortions.

**Figure 8 sensors-19-01942-f008:**
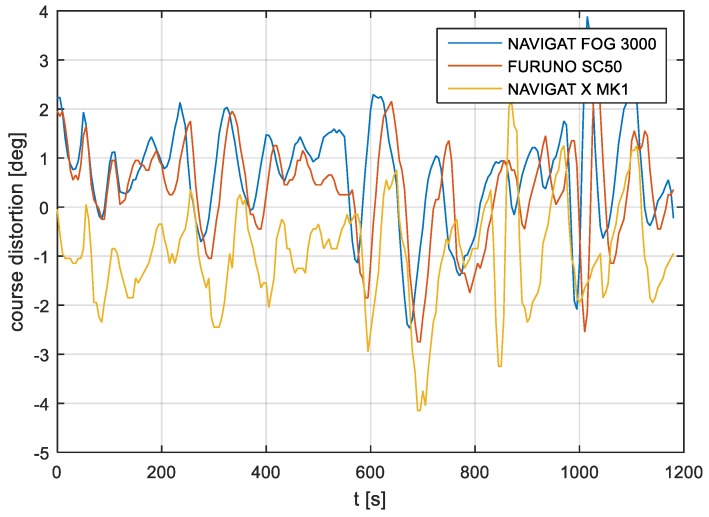
Heading recording No. 7: Course distortions.

**Figure 9 sensors-19-01942-f009:**
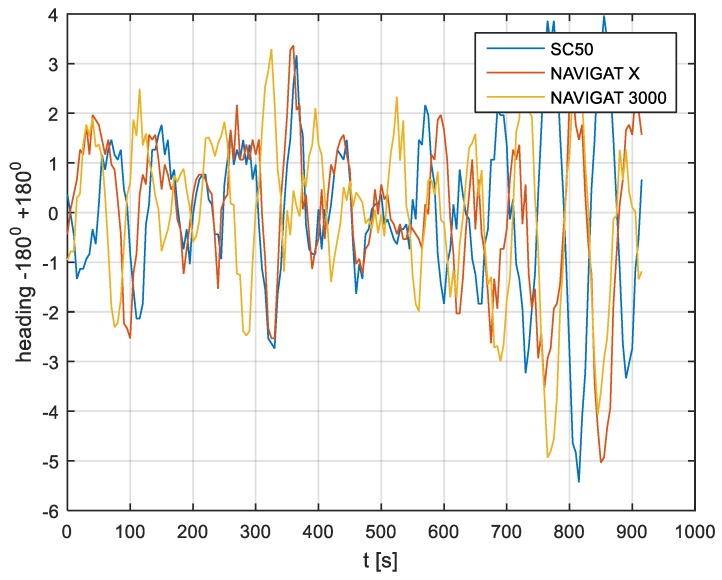
Heading recording No. 4: Post-processing data with the heading reduction (000 deg.).

**Figure 10 sensors-19-01942-f010:**
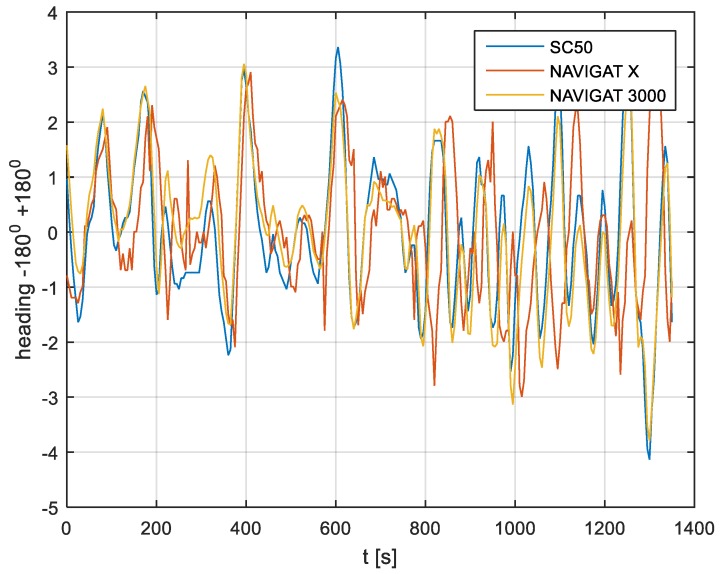
Heading recording No. 6: Post-processing data with the heading reduction (000 deg.).

**Figure 11 sensors-19-01942-f011:**
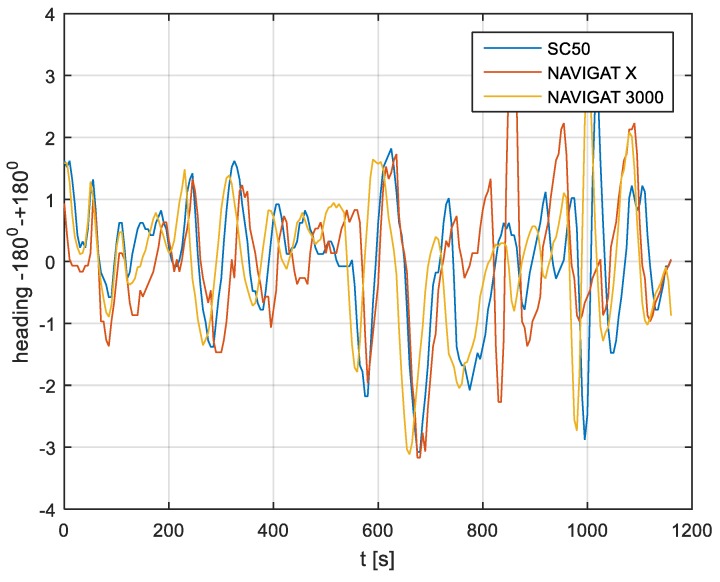
Heading recording No. 7: Post-processing data with the heading reduction (000 deg.).

**Figure 12 sensors-19-01942-f012:**
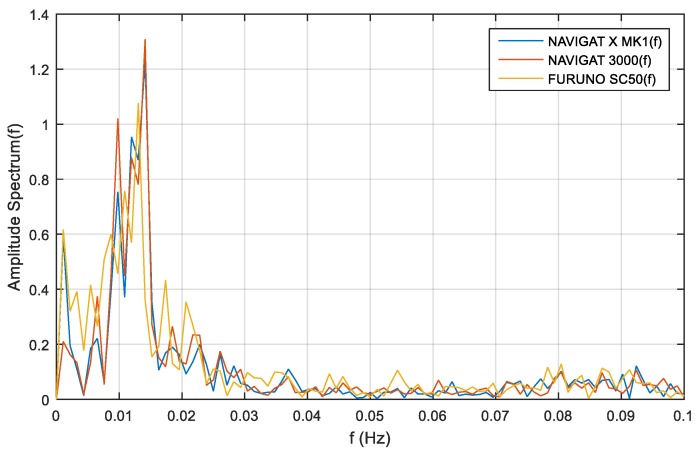
Heading recording No. 4: Single-sided amplitude spectrum of NAVIGAT FOG 3000, NAVIGAT X MK1, and FURUNO SC50 course distortions, determined using the fast Fourier transform algorithm.

**Figure 13 sensors-19-01942-f013:**
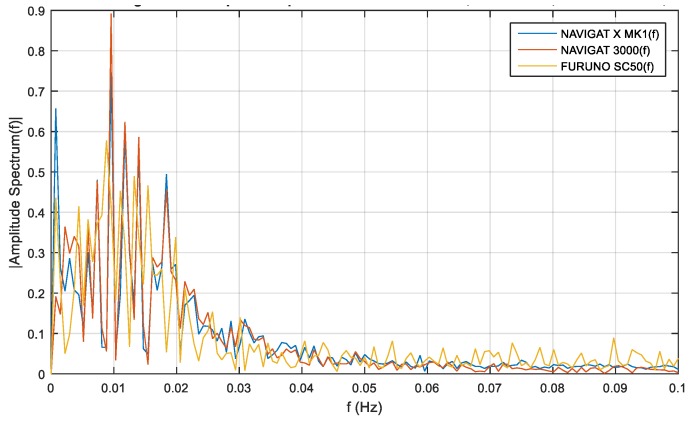
Heading recording No. 6: Single-sided amplitude spectrum of NAVIGAT FOG 3000, NAVIGAT X MK1, and FURUNO SC50 course distortions, determined using the fast Fourier transform algorithm.

**Figure 14 sensors-19-01942-f014:**
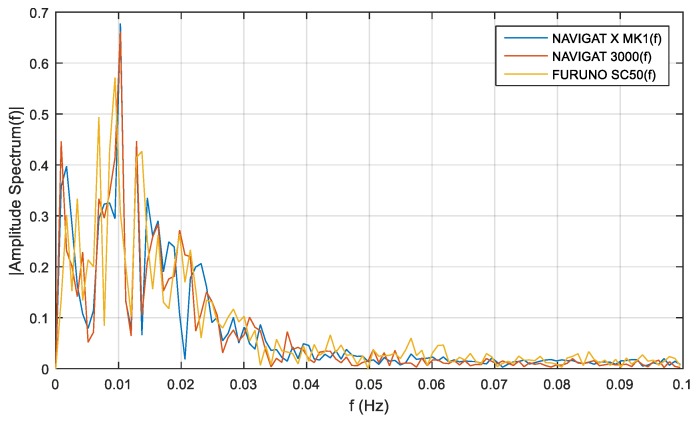
Heading recording No. 7: Single-sided amplitude spectrum of NAVIGAT FOG 3000, NAVIGAT X MK1, and FURUNO SC50 course distortions, determined using the fast Fourier transform algorithm.

**Figure 15 sensors-19-01942-f015:**
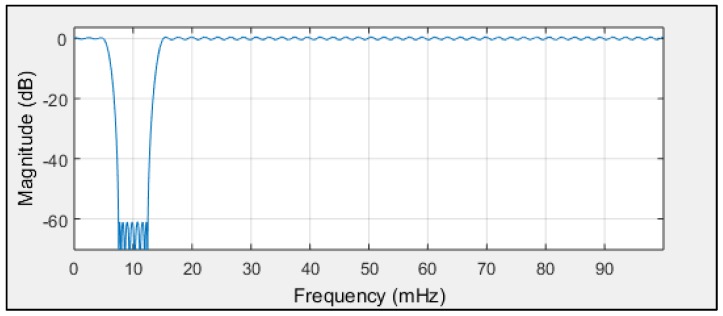
Band-stop, finite impulse response (FIR) filter to reduce low-frequency heading distortions (source: MATLAB R.2015).

**Figure 16 sensors-19-01942-f016:**
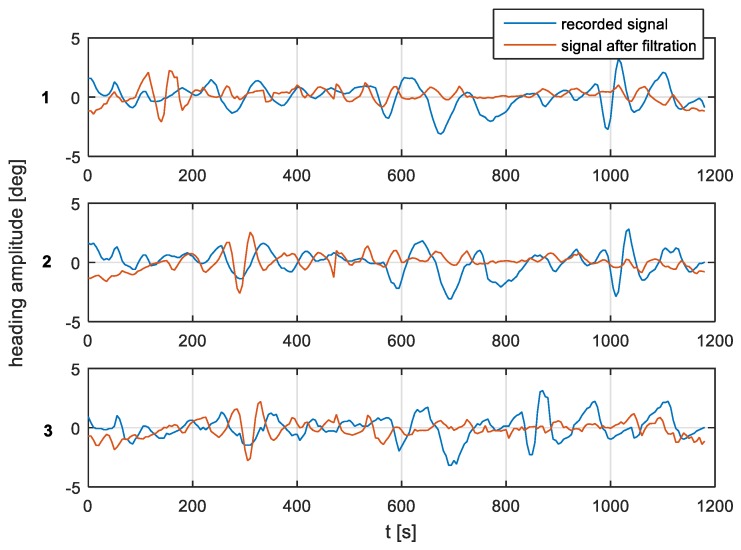
Heading recording No. 7. Heading amplitude for recorded and after filtration signal for (**1**) NAVIGAT FOG 3000; (**2**) FURUNO SC 50; and (**3**) NAVIGAT X MK1.

**Figure 17 sensors-19-01942-f017:**
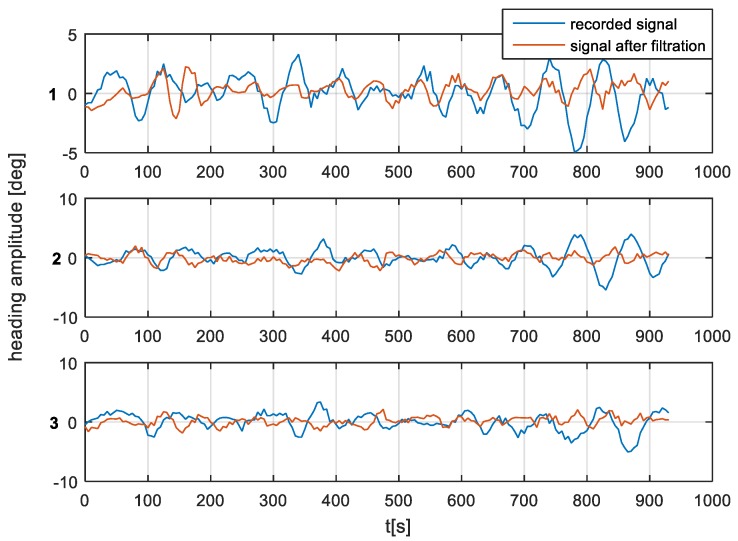
Heading recording No. 4. Heading amplitude for recorded and after filtration signal for (**1**) NAVIGAT FOG 3000; (**2**) FURUNO SC 50; and (**3**) NAVIGAT X MK1.

**Figure 18 sensors-19-01942-f018:**
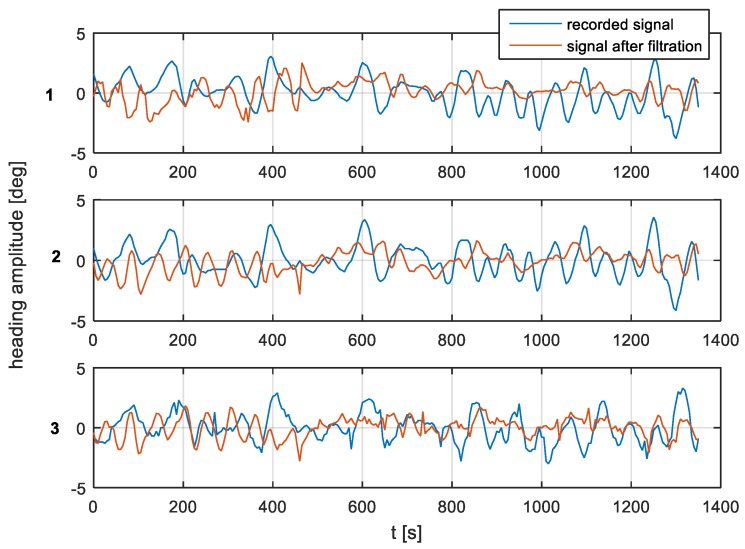
Heading recording No. 6. Heading amplitude for recorded and after filtration signal for (**1**) NAVIGAT FOG 3000; (**2**) FURUNO SC 50; and (**3**) NAVIGAT X MK1.

**Figure 19 sensors-19-01942-f019:**
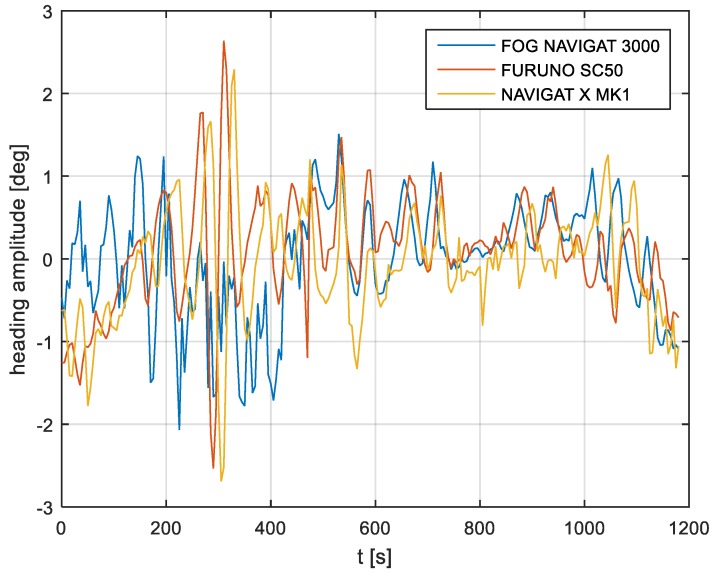
Heading recording No. 7: Heading distortions of the filtered signal.

**Figure 20 sensors-19-01942-f020:**
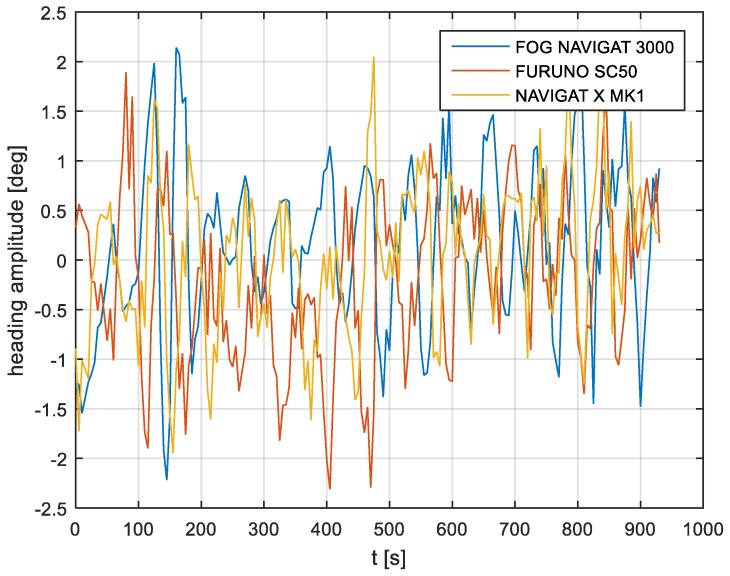
Heading recording No. 4: Heading distortions of the filtered signal.

**Figure 21 sensors-19-01942-f021:**
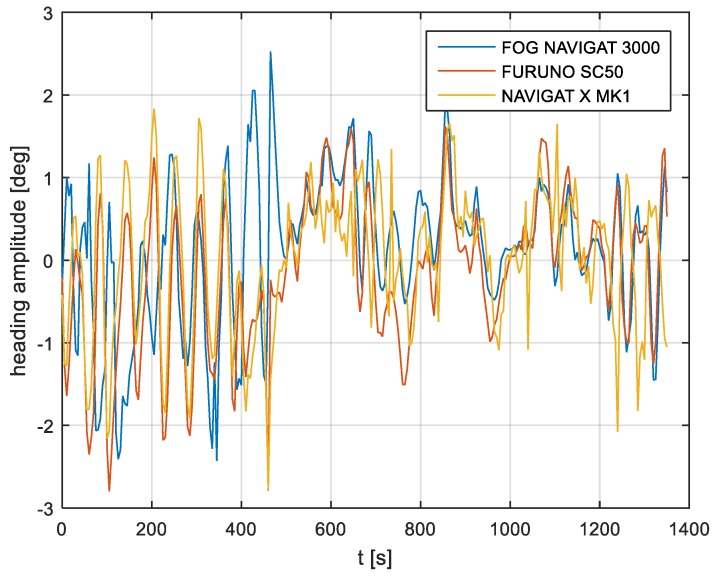
Heading recording No. 6: Heading distortions of the filtered signal.

**Table 1 sensors-19-01942-t001:** Mean errors of investigated compass systems.

The Type of Compass	Mean Error [deg.]
NAVIGAT FOG 3000	1.4
FURUNO SC50	0.9
NAVIGAT X MK1	0.7

**Table 2 sensors-19-01942-t002:** Absolute deviation outcomes of compass heading for test No. 4, 6, and 7.

Test Number	NAVIGAT FOG 3000 [deg.]	FURUNO SC 50 [deg.]	NAVIGAT X MK1 [deg.]
Test No. 4	1.53	1.23	1.30
Test No. 6	1.34	1.30	1.00
Test No. 7	1.02	0.88	1.18

**Table 3 sensors-19-01942-t003:** Compass heading mean distortion outcomes with the use of a filtered signal for heading recording No. 4, 6, and 7.

Test Number	NAVIGAT FOG 3000 [deg.]	FURUNO SC 50 [deg.]	NAVIGAT X MK1 [deg.]
Test No. 4	−0.10	0.01	−0.15
Test No. 6	0.13	−0.15	0.02
Test No. 7	0.14	0.01	−0.15

**Table 4 sensors-19-01942-t004:** Compass heading mean error outcomes with the use of a filtered signal for heading recording No. 4, 6, and 7.

Test Number	NAVIGAT FOG 3000 [deg.]	FURUNO SC 50 [deg.]	NAVIGAT X MK1 [deg.]
Test No. 4	0.54	0.52	0.52
Test No. 6	0.71	0.70	0.68
Test No. 7	0.48	0.51	0.53

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
