# Peer review of "The Compass Error Comparison of an Onboard Standard Gyrocompass, Fiber-Optic Gyrocompass (FOG) and Satellite Compass"

_sensors, 2019, doi:10.3390/s19081942_

Round 1

Reviewer 1 Report

The please kindly polish the writing, especially on the first page. There are a few grammar error. 

Instead of presenting the raw measurement data in Figure 3, 4, and 5 without much explanation, can you comment the significance of these data? What's the message the authors want to deliver?

Similar comments to later on plots. It is understandable the authors want to present the results (pre or post processing) for comparison. But the authors should comment on the significance of the data, explain the difference, and within/beyond expectation. 

Please explain how was the bandwidth of the BP filter being selected?

Author Response

Dear the reviewer,

We are very grateful to your comments and suggestions. Here are our responses to the reviewer’s comments and suggestions point-by-point.

REVIEWER 1

COMMENTS: English language and style (x) Moderate English changes required. The please kindly polish the writing, especially on the first page. There are a few grammar error.

Responses: Thank you very much for your kind suggestion, which is valuable in improving the quality of our manuscript. We correct all writing and grammar errors.

COMMENTS: Instead of presenting the raw measurement data in Figure 3, 4, and 5 without much explanation, can you comment the significance of these data? What's the message the authors want to deliver?

Responses: Thank you very much for your kind suggestion, we add the text:

These examples show perceptible resemblances of the oscillation in all indications of three compasses, surely this is the result of ship’s yaw, which is especially visible in Figure 4 and 5. However in Figure 5 can be observed the gradual approach of values shown by NAVIGAT X, to values shown by remaining. This is the characteristic behavior of the classical gyro after the ship’s turn and gets out of her dynamic (Schuler period) properties.

It is proper also to notice, that on the outline 4 and 5 the satellite compass and FOG indications are very analogous, even regarding the phase, while in Figure 3 can be observed phase shift in SC 50 indications in relation to remaining. Differences reach even to 3 degrees, while the each one accuracy should amount approx. 1 degree. During the experiment the ship swam with meridional (E and W) courses, so the speed deviation, as possible additional source of error did not appear.

We add chapter 3.2. The model of the band-stop filter with the finite impulse response (FIR). According to filter model assumptions, and research scores of Single-Sided Amplitude Spectrum presented in Figure 12, 13, 14 the bandwidth from f(pass1)=5 mHz to f(pass2)=15 mHz will be attenuated and other frequencies will be passed. Band-stop FIR was proposed to eliminate the helmsman distortions. The helmsman disturbances frequency cannot be measured directly, that is why filter parameters were estimated according to the systems accuracy investigation and the spectrum analyzed results.

In Figure 3, 4, 5 raw data measurements were presented. The measurements were carried out for various headings in accordance with the hydrographic survey profiles.

COMMENTS: Similar comments to later on plots. It is understandable the authors want to present the results (pre or post processing) for comparison. But the authors should comment on the significance of the data, explain the difference, and within/beyond expectation.

Responses:  Investigations has been conducted during bathymetric works executed by the survey ship ORP “HEWELIUSZ”. Measurement took place on directions nearly E and W, what was profitable for the diminution of errors of all instruments. Examples of the registration (Figures: 3, 4, 5) prove that the direction of movement did not have the importance. It proves also that there are no essential differences in the accuracy of examined compasses, uncertainty (errors) are different between consecutive tests (around 0.3 deg), however differences between compasses in the same test are similar.

Clearly appears the low frequency distortions in the band of 0,01Hz < f <0,02Hz. This is related to Schuler oscillations of NAVIGAT X compass which was the source of information for helmsman and as consequences it was duplicated by the rest compasses as ship in fact moves such courses.

Analysis of courses in the time domain gives opportunity to diagnose the problem deeply and excluding helmsman impact on gyrocompasses errors. In this experiment one fixed that the uncertainty (errors) of three examined instruments was similar. It is about 1 to 1,5deg but it must be stressed, that the experiment took place at the good weather, what was a requirement of bathymetric measurements at the same time. Thanks to the ship movement on stable direction on bathymetric profiles specific methodology for calculation the reference direction has been proposed.

For more accurate analysis Fourier Transform was used. The frequency domain analysis of the ship motion distortions creates a new research opportunity connected with ship inertia, weather conditions impact, etc... The frequency analysis of compass errors constitute a useful tool to investigate a simultaneous data error comparison. Frequency analyze enables the isolation of errors according to the ship’s yawing. It appears in data recordings regardless of the compass type as self-evident, when so indeed the ship moved. The frequency analysis creates a possibility to analyze outer factors of the measurement process and provides better methodology for compass error estimation. This method could be the most important in errors estimating of navigational devices. With the use of band-stop finite impulse response filter, it is possible to filter out the harmonic component resulting from long period helmsman's disturbances and ship inertia possibilities.

COMMENTS:  Please explain how was the bandwidth of the BP filter being selected?

Responses: The parameters of the BP filter was selected as follows:

f(s)=0.2 Hz - (sampling frequency),

f(pass1)=0.005 Hz - (First Passband Frequency),

f(stop1)=0.0075 Hz - (First Stopband Frequency),

f(stop2)=0.0125 Hz - (Second Stopband Frequency),

f(pass2)=0.015 Hz - (Second Passband Frequency).

The heading of the helmsman is difficult to estimate, that is why some assumptions has to be made. The bandwidth of the BP filter was selected based on analysis of the tested systems accuracy and the spectrum analyze results.

Clearly appears the low frequency distortions in the band of 0,01Hz < f <0,02Hz. This is related to Schuler oscillations of NAVIGAT X compass which was the source of information for helmsman and as consequences it was duplicated by the rest compasses as ship in fact moves such courses.

Analysis of courses in the time domain gives opportunity to diagnose the problem deeply and excluding helmsman impact on gyrocompasses errors. In this experiment one fixed that the uncertainty (errors) of three examined instruments was similar. It is about 1 to 1,5deg but it must be stressed, that the experiment took place at the good weather, what was a requirement of bathymetric measurements at the same time. Thanks to the ship movement on stable direction on bathymetric profiles specific methodology for calculation the reference direction has been proposed.

For more accurate analysis Fourier Transform was used. The frequency domain analysis of the ship motion distortions creates a new research opportunity connected with ship inertia, weather conditions impact, etc.. The frequency analysis of compass errors constitute a useful tool to investigate a simultaneous data error comparison. Frequency analyze enables the isolation of errors according to the ship’s yawing. It appears in data recordings regardless of the compass type as self-evident, when so indeed the ship moved. The frequency analysis creates a possibility to analyze outer factors of the measurement process and provides better methodology for compass error estimation. This method could be the most important in errors estimating of navigational devices. With the use of band-stop finite impulse response filter, it is possible to filter out the harmonic component resulting from long period helmsman's disturbances and ship inertia possibilities.

Should you have any questions, please contact us without hesitate.

Best regards,

Andrzej Felski,

Krzysztof Jaskólski.

Reviewer 2 Report

Dear Authors,

comments as follows:

The text is full of typos and textual errors. Please revise thoroughly;

Authors are strongly advised to prepare the text in accordance with instructions for authors;

The methodology is not written in an appropriate way, the main elements are missing;

The literature review is poor. The section containing the previous research should be more comprehensive, with papers dealing with the topic that are from relatively recent dates -  the current literature is a bit too old; 

The concluding chapter is missing (Conclusion). It is not clear what was the purpose of the paper, i.e. what are the contributions. 

Author Response

Dear the reviewer,

We are very grateful to your comments and suggestions. Here are our responses to the reviewer’s comments and suggestions point-by-point.

COMMENTS: The text is full of typos and textual errors. Please revise thoroughly;

Responses: Thank you very much for your kind suggestion, which is valuable in improving the quality of our manuscript

COMMENTS: Authors are strongly advised to prepare the text in accordance with instructions for authors;

Responses: Thank you very much for your kind suggestion, which is valuable in improving the quality of our manuscript

COMMENTS: The methodology is not written in an appropriate way, the main elements are missing;

Responses: Thank you very much for your kind suggestion, which is valuable in improving the quality of our manuscript. We add additional information in all types of article sections.

COMMENTS: The literature review is poor. The section containing the previous research should be more comprehensive, with papers dealing with the topic that are from relatively recent dates -  the current literature is a bit too old;

Responses: Thank you very much for your kind suggestion, which is valuable in improving the quality of our manuscript

We add 3 publications:

1.          Spielvogel, AR; Whitcomb, LL. Preliminary Results with a Low-Cost Fiber-Optic Gyrocompass System. Proceedings of OCEANS MTS / IEEE Conference, Washington, Oct 19-22, 2015.

2.          Johnson, BR.; Cabuz, E.; French, HB.; Supino, R. Development of a MEMS Gyroscope for Northfinding Applications, Proceedings of IEEE-ION Position Location and Navigation Symposium. Palm Springs, CA, MAY 04-06, 2010.

3.          Basterretxea-Iribar, I.; Sotes, I.; Uriarte, JI. Towards an Improvement of Magnetic Compass Accuracy and Adjustment. Journal of Navigation 2015 Vol. 69 (6), pp. 1325-1340.

COMMENTS: The concluding chapter is missing (Conclusion). It is not clear what was the purpose of the paper, i.e. what are the contributions.

Responses: Thank you very much for your kind suggestion, which is valuable in improving the quality of our manuscript. Although the conclusion section is not mandatory according to the template, the information was added to the discussion in the former manuscript.

CONCLUSION: This paper discuses specify of different ships compasses. On the basis of parallel recordings of data from three different devices installed on the same ship authors investigate uncertainty of measurements in the amplitude and frequency context. In the real experiment it was checked that indications of the price of different compasses had differed insignificantly.

The frequency analysis of compass errors constitutes a useful tool to investigate a simultaneous data error comparison. It was demonstrated, that frequency analyze enables the isolation of errors produced because to the ship’s yawing. The model of the band-stop FIR is a good tool to eliminate low frequencies heading distortions. According to filter model assumptions, and research scores of Single-Sided Amplitude Spectrum the bandwidth from f(pass1)=5 mHz to f(pass2)=15 mHz will be attenuated and other frequencies will be passed. Band-stop FIR was proposed to eliminate the helmsman distortions. After band-stop FIR usage, the accuracy of the 3 types of compass systems seems to be comparable. During the experiment the ship was steered according to information from NAVIGAT X classical gyrocompass and in each registration exists low frequency component, probably being descended from dynamic properties of this compass. However interesting is the relation of dynamic properties of the ship to this properties of gyrocompass. Simply – if the source of information about the heading will be different compass, without of this low frequency oscillation – how will be remain the ship? This could be the direction of future investigation.

Should you have any questions, please contact us without hesitate.

Best regards,

Andrzej Felski,

Krzysztof Jaskólski.

Paweł Piskur

Round 2

Reviewer 2 Report

Dear authors, 

Thank You for considering the remarks. I believe that in this form the manuscript can be considered for publication.

With kindest regards